# Unmanned Aerial Vehicles for Debris Survey in Coastal Areas: Long-Term Monitoring Programme to Study Spatial and Temporal Accumulation of the Dynamics of Beached Marine Litter

**Silvia Merlino [1],\*** , **Marco Paterni [2]**, **Andrea Berton [2]** and **Luciano Massetti [3]**

1   Istituto di Scienze Marine del Consiglio Nazionale delle Ricerche, ISMAR – CNR, 19032 Lerici (SP), Italy
2   Istituto di Fisiologia Clinica del Consiglio Nazionale delle Ricerche, IFC – CNR, 56124 Pisa (PI), Italy; marco.paterni@ifc.cnr.it (M.P.); andrea.berton@ifc.cnr.it (A.B.)
3   Istituto per la Bioeconomia del Consiglio Nazionale delle Ricerche, IBE – CNR, 50145 Firenze (FI), Italy; luciano.massetti@ibe.cnr.it
\*   Correspondence: silvia.merlino@sp.ismar.cnr.it; Tel.: +39-0187-178-8902

**Abstract:** Unmanned aerial vehicles (UAVs) are becoming increasingly accessible tools with widespread use as environmental monitoring systems. They can be used for anthropogenic marine debris survey, a recently growing research field. In fact, while the increasing efforts for offshore investigations lead to a considerable collection of data on this type of pollution in the open sea, there is still little knowledge of the materials deposited along the coasts and the mechanism that leads to their accumulation pattern. UAVs can be effective in bridging this gap by increasing the amount of data acquired to study coastal deposits, while also limiting the anthropogenic impact in protected areas. In this study, UAVs have been used to acquire geo-referenced RGB images in a selected zone of a protected marine area (the Migliarino, Massacciuccoli, and San Rossore park near Pisa, Italy), during a long-term (ten months) monitoring programme. A post processing system based on visual interpretation of the images allows the localization and identification of the anthropogenic marine debris within the scanned area, and the estimation of their spatial and temporal distribution in different zones of the beach. These results provide an opportunity to investigate the dynamics of accumulation over time, suggesting that our approach might be appropriate for monitoring and collecting such data in isolated, and especially in protected, areas with significant benefits for different types of stakeholders.

**Keywords:** unmanned-aerial-vehicles; UAVs; anthropogenic-marine-debris; AMD; beached-marine-litter; BML; marine-protected-areas; MPA; ortho-photo; marine-pollution; accumulation-rate

---

## 1. Introduction

Interactions between geosphere and anthroposphere, in sensitive areas such as the land–sea interface, are constantly evolving due to population growth and exploitation of natural resources. Therefore, the growing problem of the accumulation of anthropogenic marine debris (AMDs, or marine litter–ML), especially in isolated/protected coastal areas, is one of the emerging problems of recent decades. The interest in AMDs pollution in recent years has led to a significant increase in data related to such material in oceans [1]. In the Mediterranean area, the increasing knowledge of the concentration and type of ML [2–7] and the increasing efforts to survey off-shore areas, have not been accompanied by an equally increasing knowledge of the sources, composition and distribution of materials deposited along the coast (beached marine litter, BML), and the mechanism through which they accumulate in

particular coastal areas. A long stay on the coast of BML can cause considerable damage. Especially plastic objects left on the beach for months/years are subject to photodegradation at a higher rate than expected if they were at sea [8,9]. Rapidly fragmented, reduced to meso-plastics (5 mm–2.5 cm) and micro-plastics (MPs, ≤ 5 mm long), they mix with the substrate and can produce a stream of particles flowing into the sea [10–12] adding to those directly released by rivers, which are key agents in the release of macro and micro marine litter in oceans [13,14]. There is an urgent need to develop new methods of spatial and temporal mapping of beaches to identify the areas of greatest accumulation, quantify the abundance and types of material, and trace their origin, in line with the protocol and standard monitoring strategies [15–19]. Numerous studies have reported that 80% of waste present in the sea is probably of terrestrial origin and rivers also seem to play a key role in the transport of debris from land to oceans [20]. Therefore, it becomes important to estimate the flow of material transported by water courses and its impact on the areas surrounding the river mouths. So far, few studies have investigated the transport, deposition, and accumulation of AMDs through internal water. To better understand this problem, monitoring actions are needed to verify how rivers transport AMDs and how they affect coastal deposits. These surveys should collect data in a consistent manner throughout the investigated territory, in one or more seasons, and with different replicas, trying to correlate the data obtained with the anthropic impact (urbanization, presence of parks and protected areas) and with the morphological characteristics (rivers, ports, types of beaches) of the area. Particularly at risk are the MPAs, which often suffer from a large influx of AMDs, as they are located in or near densely populated and industrialized areas. AMDs in protected coastal areas are often difficult to clean from waste due to the inherent difficulty of reaching these isolated areas that are not served by roads or facilities, and also due to regulations that limit human intervention. In this context, the use of aerial survey could be a valuable aid. To get the best results from aerial survey processing it is important to choose the right scale [21]. Since the highest resolution of commercial satellite images is about 0.3 m (WorldView-4), this platform is not the most suitable for observing beach waste [22]—a spatial resolution below decimeter is required and UAVs, especially commercial UAVs, have proven to be effective in this respect. The longer battery life, the ability to plan automatic flights with easy-to-use ground station software, and their small size are real advantages, and the structure from motion algorithms (SFM) allow accurate digital elevation models (DEM) and ortho-mosaic terrain models over large areas. Today UAVs are increasingly accessible and have widespread applications, such as in environmental monitoring systems for agroforestry, structural geology, archaeology, marine habitats, supervised hazards, and accidents [23–39], and recently also in monitoring ML on the coast [40–44] or that floating in rivers [45]. These studies are not uniform with regard to the data processing procedures, ranging from visual interpretation of images [42] and analysis of the spectral profile of litter [46], to the use of machine learning methods [43,44]. Moreover, since this is an "emerging field of study", there is no single standardized protocol for data acquisition and processing, but only a few suggested protocols [42,43]. The difficulty of developing scalable procedures that do not depend on local environmental constraints, is also due to the different objectives to be achieved. In any case, most of the studies carried out focused on the detection of BML stocks, especially in isolated areas. The advantages offered by UAVs, in terms of survey resolution and repeatability, are particularly suitable for the purpose we are interested in, that is to study the model of aggregation and distribution of BML in such remote areas, and are particularly useful to monitor the most sensitive areas, such as protected areas. So far, little attention has been paid to the long-term study of a particular area in order to understand the dynamics of BML deposition, and to obtain the rate of accumulation and the variability of spatial distribution over time. The vertical spatial distribution (cross shore) of debris on a beach has its own dynamics, which is strongly influenced by the physical processes determined by the wind and waves on the beach profile. Therefore, to understand this phenomenon, it is important to monitor it over a long period, with frequent sampling [46,47]. The "manual" collections and cataloguing of BML are the usual way of carrying out such monitoring, but they take time and involve many people. UAVs can reduce both monitoring time and human staffing requirements. For this reason, starting

from April 2019, we have implemented a pilot monitoring program through UAVs in the Migliarino, Massacciuccoli, and San Rossore (SRPRK) park, a marine protected area with 34 km of protected coastline, north of the mouth of the Arno river. We planned to use UAVs to acquire geo-referenced RGB images in a selected area about 100 meters long, from the dune crest to the low-tide terrace (shoreline base), during a one-year monitoring program (about two recognition flights per month). For each monitoring and image acquisition date, a post-processing system allowed the localization and identification of BML within the area scanned by ortho-photos. A specially created software for pattern recognition (based on visual interpretation of the images, [42]) provided the estimation of typology, quantity, density, and position of the identified items (see Material and Methods).

The UAVs used in this study were in the 'multicopter' category, and in the 'light' and 'very light' classes, and had an autonomy of about 30 minutes of flight, sufficient for the purposes of our current survey, as suggested in previous studies [42,43]. In our case, each flight allowed us to cover the entire selected area (100 m × 15 m), even considering that it used a conservative approach with high overlap between images and a "stop and go" shooting method that increased the overall flight time.

In this specific area, where there are airworthiness constraints due to the presence of a control traffic region (CTR), flight missions were allowed at specific heights provided for by the regulations; in any case, a resolution of at least 2/2.5 cm/pixel (and even higher) would be guaranteed, which would be sufficient to recognize even the cotton buds or caps, i.e., the small BML typically present on beaches.

## 2. Materials and Methods

### 2.1. The Study Area

The target area was the afitoic backshore of a stretch of sandy beach inside the marine protected area of SRPRK. This area, located between the two rivers Arno (N 43°40'47.408", E 10°16'40.466") and Serchio (N 43°47'1.704", E 10°16'0.016") is affected by the marine current that goes from the mouth of Arno to the north, with a considerable transport of fluvial material. The Arno River is, in fact, an important Italian water course that crosses the Tuscany region, running through large cities like Florence and Pisa, and industrial and production centers such as the province of Prato and Pontedera. The limitation to tourism in this area of the Park allows the study of the dispersion of marine debris and its accumulation on the coasts, caused by natural and meteorological events, as there is no direct contribution to such accumulation by human presence. This site is in fact located within the area "A" of SRPRK, which means there is an absence of tourism throughout the year. Access to this area is forbidden from both land and sea, and access is only allowed for research purposes. In summer, some excursions are made with environmental guides, but only on a few specific paths and never beyond them. The beach is a "natural" beach, with a dune cordon parallel to the coastline that delimits the hinterland. In our case the foreshore is very small, given the weakness of the tidal phenomena, as on most of the coasts of the Italian peninsula. We have taken into account the maximum extent of the tide in this area when choosing the points of our stretch of beach (10°16'40.70" E 43°42'55.07" N; 10°16'42.12" E 43°42'51.86" N; 10°16'41.68" E 43°42'51.74" N; 10°16'40.23" N. E 43°42'54.93" N), such that it started at the edge of the "swash zone" (wave run-off zone). The size of the selected area was about 100 meters long and 15 meters wide, with a south-west exposure (Figure 1).

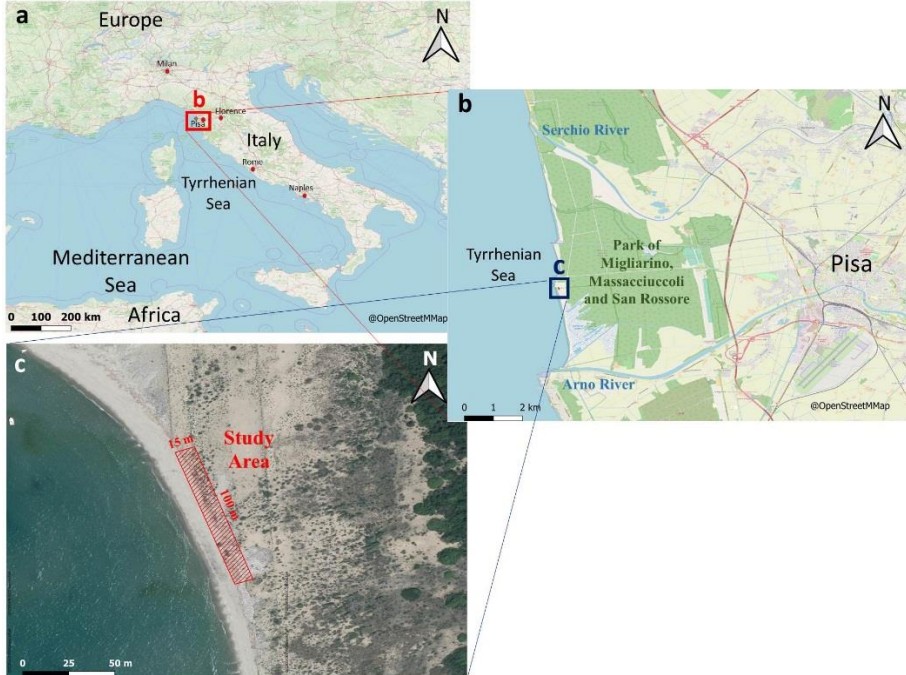

**Figure 1.** Geographical location of the stretch of beach studied, on the Northern Italian coast (coordinates: 10°16'40.70″ E 43°42'55.07″ N; 10°16'42.12″ E 43°42'51.86″ N; 43°42'51.86″ N; 10°16'41.68″ E 43°42'51.74″ N; 10°16'40.23″E 43°42'54.93″N). Maps created by using QGIS 3.12 [48], @OpenStreetMap Contributors [49] and Geoscopio WMS service by Regione Toscana [50].

## 2.2. Characteristics of the Used UAV

In the present study the Phantom 4 PRO v2 quadcopter [51] was used. It is a commercial UAV suitable for this type of application, thanks to the good resolution of the camera (5472 × 3078 pixels, which, flying at 6 meters above ground level allowed us to reach the theoretical value of 0.16 cm/pixel—with perfectly flat ground and in the best conditions—and, in our case of not perfectly flat ground, 0.18 cm/pixel), the compactness of the aircraft, and flight stability. It had a titanium and magnesium alloy structure, increasing the strength of its frame and reducing its weight; together with the good battery capacity (5870 mA), this gave a flight time of up to approximately 30 minutes. It had a gimbal three-axis stabilized camera with a 1-inch 20-megapixel CMOS sensor (Figure 2), capable of shooting up to 4K/60 fps video and photo bursts, at up to 14 fps. The gimbal was set to −90° to look at nadir. This allowed it to capture photos perpendicular to the direction of flight. It was equipped with HD video transmission capable of reaching a maximum range of 7 km. The correct position management was obtained thanks to two satellite tracking systems: GPS and GLONASS. The use of UAVs for 3D mapping of the terrain or sites has the advantage to access utilities, like waypoint mapping for identifying the surveyed area and flight path planning and control, provided by third-party applications. Three sets of dual vision sensors formed a 6-camera navigation system that worked constantly to calculate the relative speed and distance between the UAV and any object; this system allowed it to fly more safely and avoid obstacles along the way. A remote controller allowed a pilot to control the flight of the UAV; a smartphone (or tablet) could be connected to the remote controller to view the camera, read the telemetry, and enable automatic functions. The maximum speed was 72 km/h and the maximum control range was 7 km from the driver. The UAV operated automatically using the Drone Harmony (DH) ground station software (see Section 2.4). The mapping and modeling of the aerial photography area was selected during the configuration process, and the flight plans were selected. The automatic process of the mission included take-off and landing, route planning and calculation of the corresponding spatial resolution of the flight altitude, which were displayed on the screen. Establishment of flight altitude depended on the spatial resolution we wanted. As the

number of pixel per item depended on the type of object (our test gave 20–30 pixel for a bottle cup, 18–22 pixel for cotton-buds, and 40–50 pixel for a spoon) with best light condition and since these conditions were not always present, we tried to increase the resolution by acting on the flight altitude, so as to guarantee at least twice the pixel/item values previously tested.

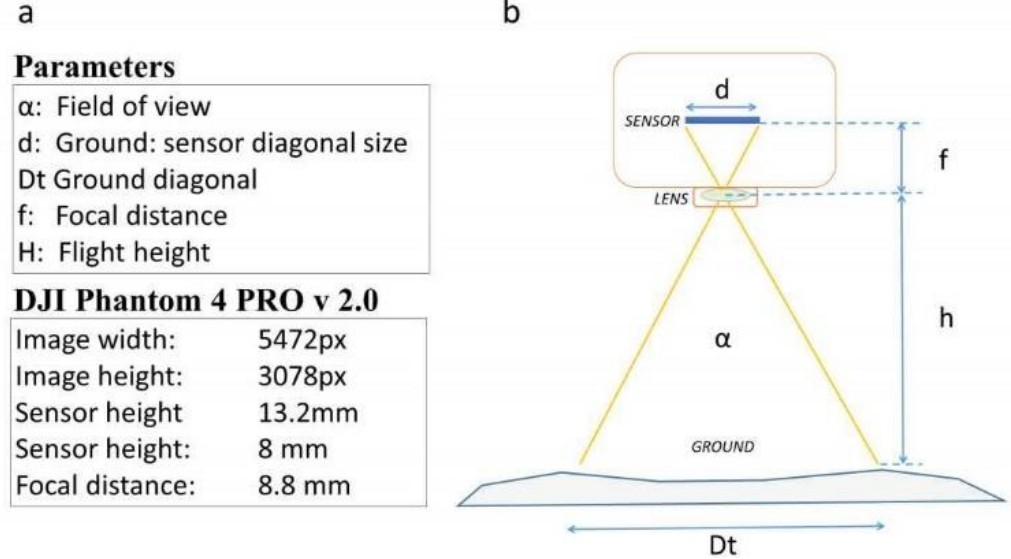

**Figure 2.** (**a**) Sensor characteristics of the DJI Phantom 4 PRO v 2.0; (**b**) parameters for calculating the distance of the sample to the ground: sensor diagonal size (**d**), focal distance (**f**), field of view ($\alpha$), ground diagonal (Dt), and vertical ground distance or flight height (**h**). With this choice, a spatial resolution of 0.01–0.03 m was obtained, which was sufficient to recognize the smallest macro beached marine litter (BML).

Once the flight altitude of the UAV was established, the ground sample distance (GSD) was calculated using the parameters in Figure 1 and Equations (1), (2), and (3):

$$\alpha = 2 \times \text{atan}(d2/f), \tag{1}$$

where $\alpha$ is the field of the view angle, d2 is half the diagonal of the sensor area, and f is the focal distance,

$$Dt = h \times \tan(\alpha/2) \times 2, \tag{2}$$

where Dt is the ground diagonal, h is the vertical ground distance, and $\alpha$ is the field of view angle,

$$GSD = Dt/n, \tag{3}$$

where GSD is the ground sample distance, Dt is ground diagonal, and n is number of pixels.

## 2.3. Survey Realization

After having identified and delimited the coastal profile of the study (a stretch of about 100 meters in length and 15 meters in depth, starting from the coastline to the dune area), we established the flight plans and the take-off/landing points for the UAV (Figure 3). These plans must have guaranteed the necessary resolution, the respect of the autonomy of the aircraft, the compliance with the regulations in force, and the acquisition of photos with overlap of at least 70%–80%. This overlap level ensured a greater precision during the preparation of the ortho-mosaic. In fact, by using a low altitude flight, the scanning area for each photogram was limited (8.76 m × 5.84 m) and an error of even 1 or 2 meters could have had a big impact on the photogrammetric reconstruction. A high overlap, therefore, compensated for this fact, and also allowed us to get a better 3D reconstruction.

UAVs operating in the multicopter category and in the "very light" classes had an autonomy of about 25/30 minutes of flight. In order to have the maximum resolution in the photos, a flight altitude of 6 m was chosen. Using this configuration, we got a ground sampling distance (GSD) of 0.18 cm/pixel. In addition, specific precautions were used to ensure optimal image capture: 1 m/s speed with a "stop and go" mode for each shot, to ensure shooting in a stationary position due to low flight altitude and to avoid blurred photos; with the manual focus set to infinity (i.e., autofocus disabled) to avoid variations in focus; initial exposure setting of autoexposure (AE) was disabled to avoid variations in brightness. In the specific study area, several scans were performed using UAVs. For each scan, the specific software (Agisoft Photoscan Professional) provided the alignment and creation of an orthophoto for the entire area. The memorization of the flight plans allowed the replication of the scans at a later time, keeping the same areas of interest and the established take-off/landing points. We used the Litchi application to manage the storage of the flight plan; if the conditions required landing before the end (for example due to the sudden occupation of the airspace), the Litchi application allowed you to resume the flight from the point where it was interrupted.

After the initial scan of the monitoring program (12 April 2019), the removal of anthropic material from the stretch of beach considered was carried out, excluding only objects with a linear dimension less than 2.5 cm (OSPAR protocol). The collected material was subsequently catalogued and counted according to a protocol previously adopted in this type of monitoring; this protocol integrated the Marine Strategy Framework Directives (MSFD) survey procedures [16], the OSPAR guideline [15] for size and type classification of BML and citizen science contribution, involving volunteers, researchers, and university students during beach cleaning operations, classification, and counting of objects (SeaCleaner protocol [52,53]). As a result, we could compare and match the ortho-photo data with those collected by standard manual surveys. A second survey was then carried out by acquiring images immediately after the cleaning of the beach. From this date, every 10/15 days and for a period of about 4 months, surveys were carried out with UAVs. The scanned images/videos were transferred to the servers. A post-processing system that uses a pattern recognition software, located and identified the different BML within the scanned area and estimated the accumulation rate of the different classes of objects and dimensions and other parameters of interest. On 13 July 2019, the beach was cleaned for the second time, the BML were catalogued and counted, and the procedure described was started again for the second period of study.

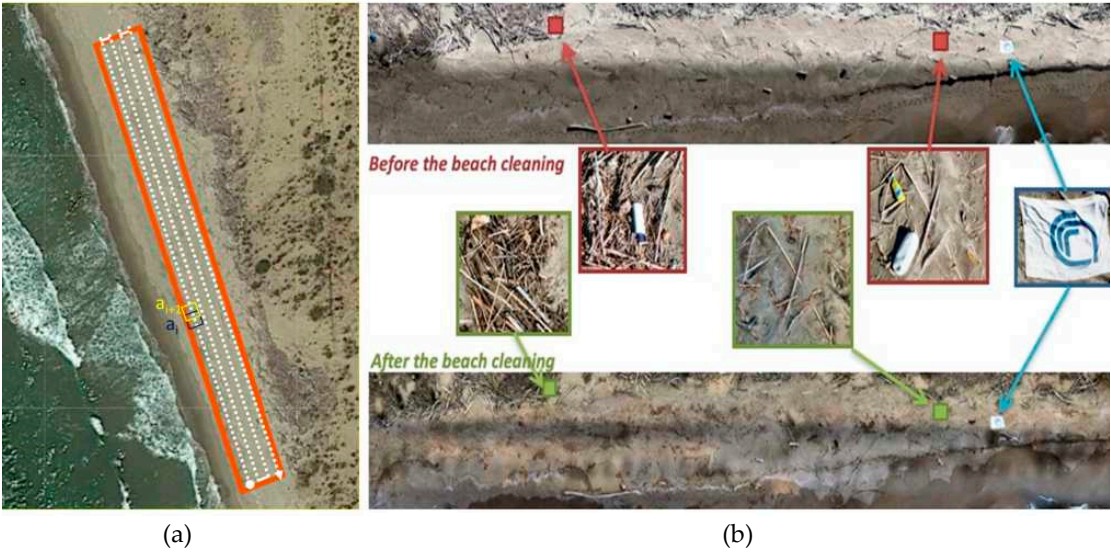

　　　　　　　　　　　(a)　　　　　　　　　　　　　　　　　　　　　　　　　　　(b)

**Figure 3.** (**a**) An example of a transect performed by unmanned aerial vehicles (UAVs) to obtain the necessary images for the entire coverage of the area of interest, with an overlapping of the shooting areas of the adjacent frames. (**b**) The stretch of beach investigated before and after the complete cleaning carried out by researchers and volunteers.

### 2.4. Image Acquisition and Processing

The software used for the automatic flight of UAVs was "DRONE Harmony", a commercial software (free to use only for one month) that can perform several operations (to create the flight plan necessary, to capture the photos, etc.) in a simple and accurate way [54]. The total acquisition time of all images of the studied area, on each date, was about 21 minutes. The speed of the flight was 2 m/s, but the total acquisition time increased because the "stop and go" mode option was used at each selected point for the photographic acquisition. The photogrammetry technique was used to define the position, shape, and size of the objects on the ground, using the information contained in appropriate photographic images of the same objects, taken from different points. The photos were in fact taken such that there was an overlap between the adjacent frames, with a coverage of about 70% or 80% (see Section 2.3). This technique can be used both at the ground level and in aerial mode, and allows the obtainment of a 3D reconstruction of the objects (Figure 4), whose potential, even if not exploited in the present study, could be useful for several applications (Supplementary Material B).

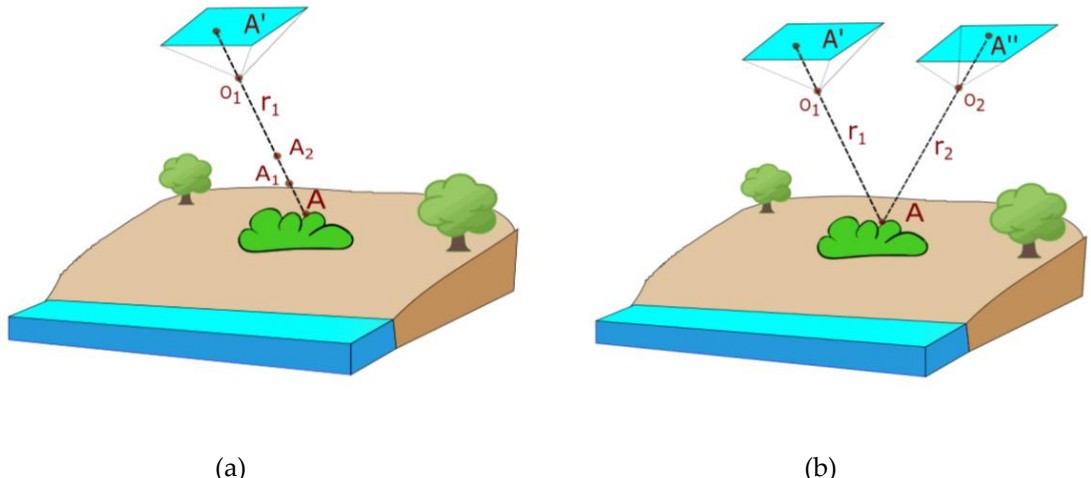

(a)　　　　　　　　　　　　　　　　　　　　　　　　　　(b)

**Figure 4.** Photogrammetry technique allowing the 3D reconstruction of the objects. By precisely knowing the position of the homologous points A' and A'' on the two photographs, and the spatial position of the two sectors and the two perspective centers O1 and O2, the point A remains geometrically defined, since it is the intersection point of the two projecting rays r1 and r2 connecting the two homologous points with the perspective centers (**b**). This does not happen with a single photo shoot (**a**).

In our specific application, we used Agisoft Photoscan, a standalone software product that performs photogrammetric processing of digital images and generates 3D spatial data for use in GIS applications, cultural heritage documentation, and visual effects production, as well as for indirect measurements of objects of various scales. The use of this software allows us to obtain ortho-mosaic, a calibrated image that constitutes the ortho-rectified mosaic of the entire area covered by the scan. We also obtain the digital elevation model (DEM), with the aim of estimating the height variations of the ground that must be used to correct the dimensional measures of the objects (see Figure 5).

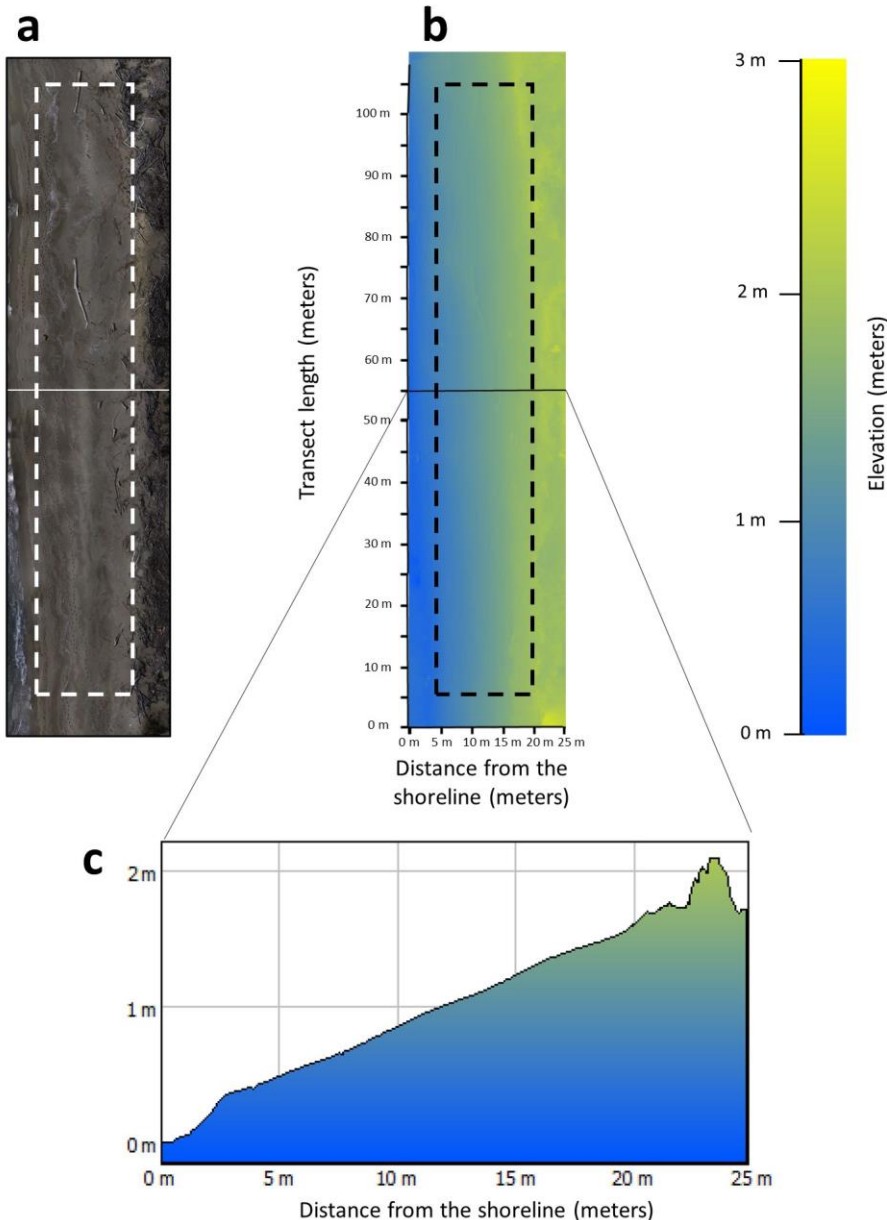

**Figure 5.** Orthophoto (**a**) and digital elevation models (DEM) (**b**) of the study area, with the profile of the beach corresponding to the central point of the sampled transect (**c**). Distances and elevation are indicated in the image, with the dotted line that delimits the study area inside the beach.

## 2.5. Data Acquisition from Images and Data Analysis

After data collection, the image sets were processed by Agisoft Photoscan for the generation of dense points cloud and Digital Terrain Models. GPS information extracted from the EXIF (EXchangeable Image File) information of each image file was used to create a georeferenced ortho-photo map (ortho-mosaic), with a resolution of 0.18 cm/pixel. Once the ortho-mosaic was obtained, it was possible to extract the data of interest for this study. We were mainly interested in obtaining an estimate of the spatial coverage of the monitored stretch of beach, over time, by the BML. This involved the knowledge of the surface occupied by the surveyed objects, and not only their possible type identification (material), the standard linear dimension and the numerical estimate [15,16]. For this reason, we decided to develop a semi-automatic software (waste mapping, WM) to quantify the waste detected with the ortho-photos acquired with aerial survey. This software (currently only available for internal use, but we plan to share it within the concerned scientific community when

we have concluded the necessary updates) was developed for the most popular operating systems (Windows, OSX, Linux); it could load an ortho-rectified image of the analyzed area and also offered analysis tools on it. The image must contain some acquisition data as ground sample distance and geographic coordinates; if not available, these data could also be entered manually. The user had a cursor which identified the presence of the object; subsequently, the user could have an automatic drawing of the shape of the object, but, if it did not work (overlapping objects, unclear image, noise, etc.), he could manually carry out the drawing; the user then associated the object with a class (plastic, metal, multi-material, etc.). Once these operations were performed on the whole image, the software estimated the number of objects, GPS position, area, and principal linear dimension of each object, and then calculated the properties of the objects: total number for each material type (plastic, glass, metal, etc.), total number for each size category, both in terms of standard linear dimensions and measured area, also expressed as a pseudo-color map. This visualization of the local density of BML (local percentage of area covered by the objects) was obtained in the following way—the whole stretch of the beach was divided into $32 \times 6$ rectangles, each measuring $3.125 \text{ m} \times 2.5 \text{ m} = 7.8125 \text{ m}^2$: = TA (see Section 3.1 for the results of this visualization methods). Then, for any rectangle of beach, the local density surface area (DA) would be DA = $\sum$ OA$_i$/TA, where OA$_i$ is the object coverage area, with i varying from 1 to n; n = total number of detected objects inside the considered rectangle.

All information obtained through the WM software could be exported to a CSV (Comma Separated Values) file and managed by other processing software. The effectiveness of this evaluation method was validated by distributing a defined number of objects of a known size (surface area and linear dimension) on the same stretch of beach (Figure 6a,b, and Figure 7c,d) and then analyzing the WM errors in identifying the sizes and the number of the same objects. Thus, we evaluated the errors in percentage terms:

- PE$_S$ = (Sm-Sa)/Sa $\times$ 100 where PE$_S$ is the percentage error on size (object surface area) estimation, Sm is the measured object size, Sa is the actual object size;
- PE$_N$ = (Nm-Na)/Na $\times$ 100 where PE$_N$ is the percentage error on the number of objects estimation, Nm is the measured object number, Na is the actual object number.

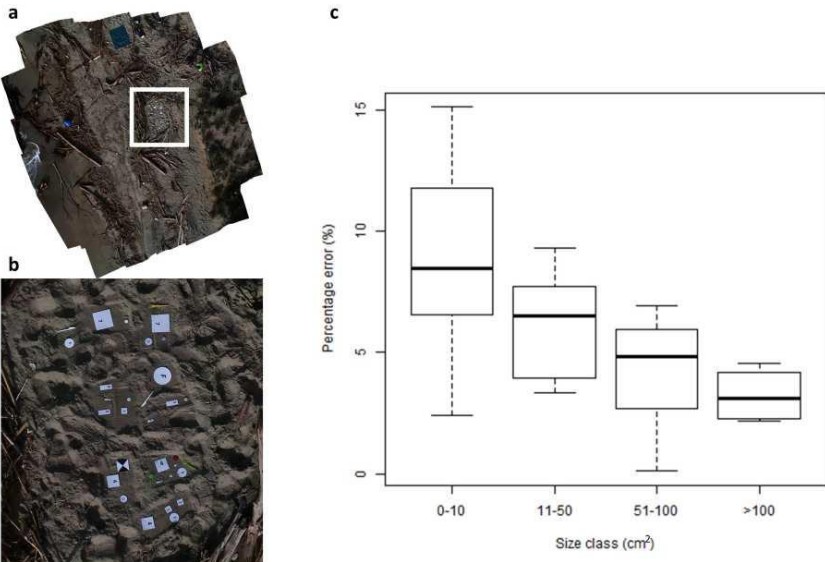

**Figure 6.** Error evaluation on items size (object surface) estimation. (**a**) Part of the whole study area; (**b**) zoomed-in view on the portion of beach used for validation, of about 18 m$^2$; (**c**) boxplot of percentage error (PE$_S$) per size class (in m$^2$). PE$_S$ was calculated using Microsoft Excel and boxplot were made with R 3.4.1 software for windows (https://www.r-project.org/). Defining PDs the "percentage of detected objects by UAVs", we obtain that PDs is 100–PE$_S$.

Concerning the object surface area estimation, in Figure 6c we can see how the relative error grows with a decreasing size class, but it is still quite small for all classes. For item counting, the relative errors were classified according to the linear size of the objects, because it was the same classification used during the standard manual monitoring (see Section 3.3). Additionally, in this case, the percentage error increased while the size class decreased (Figure 7c). The validation of this method was done in the best possible weather conditions (Figure 7a), but we also showed, in Figure 7b, how the ortho-photo image could be in the worst weather conditions (wind, change of brightness, etc.). In this case the relative error increased, especially for smaller objects.

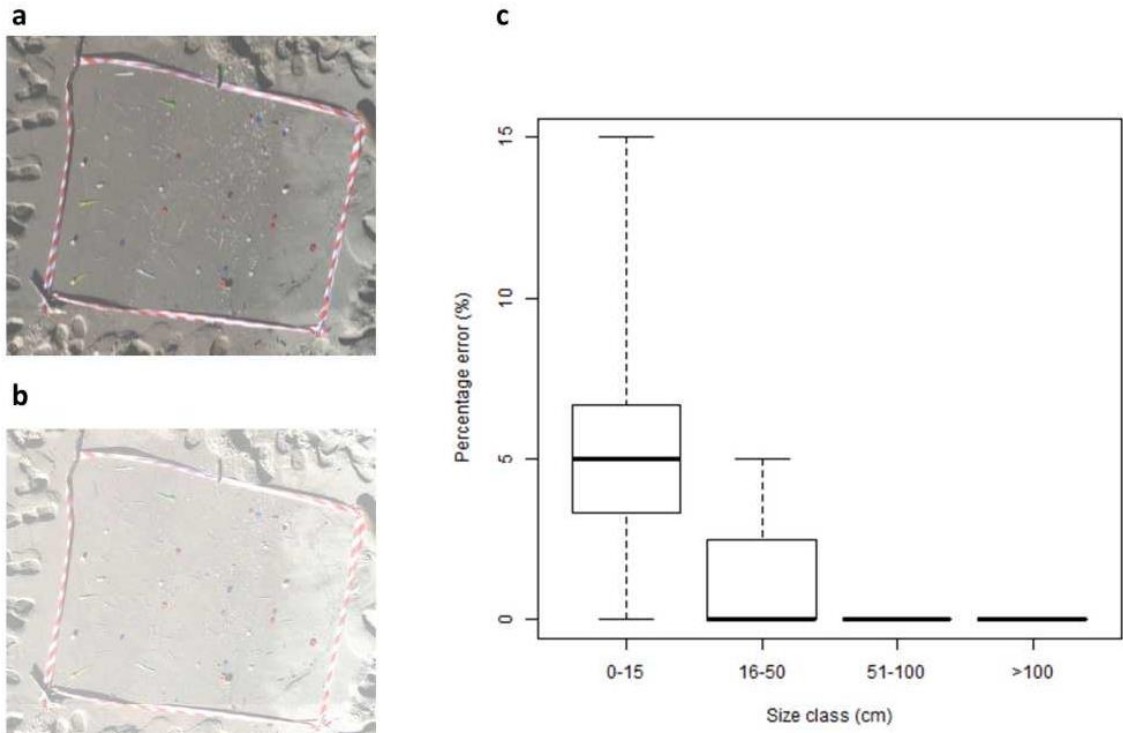

**Figure 7.** Error evaluation on items counting estimation in (**a**) good weather condition; (**b**) a case of possible bad weather/not optimal visual condition; and (**c**) boxplot of percentage error in items counting (PE$_N$) per size (linear dimension, following OSPAR guidelines prescriptions). PE$_N$ was calculated with Microsoft Excel and the boxplots were made with the R 3.4.1 software for windows (https://www.r-project.org/). PD$_N$ indicates the percentage of detected objects by UAVs; PD$_N$ = 100 − PE$_N$.

## 3. Results

From April 2019 to January 2020 we carried out 17 total recognition flights of the studied area, within the SRPRK, near Pisa, Italy. For each flight, we have elaborated and realized an ortho-mosaic. The visual screening of each ortho-mosaic took about 60–80 minutes, and was carried out using our WM software, which allowed us to obtain different types of information.

### 3.1. Two-Dimensional Distribution of BML on the Beach

WM enabled us to estimate the size of the classified objects in terms of the surface area they occupied, to visualize the amount and the distribution of BML over time in the different zones of the beach (Figures 8 and 9), and to investigate the dynamics of their accumulation over time (Figure 10). This is an aspect that, to our knowledge, has not been previously investigated using UAVs.

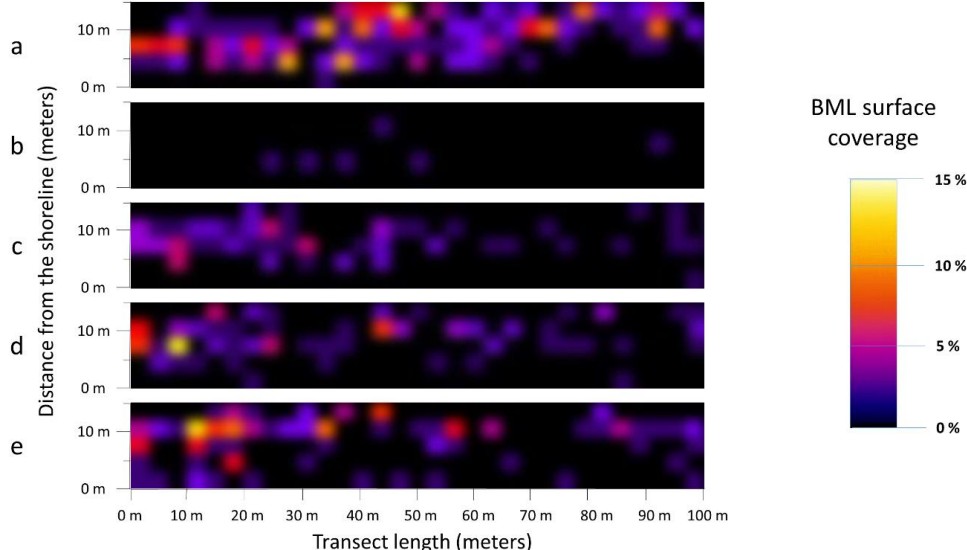

**Figure 8.** Variation in beached marine litter surface coverage of the monitored area, over time, for the first temporal period (from April 2019 to July 2019). The sequence of strips, from (**a**) to (**e**), shows the beach before the first total cleaning in (**a**), immediately after the first cleaning in (**b**), and at the following dates of our monitoring in (**c**), (**d**), and finally (**e**), after 90 days. The correspondent dates were: (**a**) 15 May 2019; (**b**) 24 May 2019; (**c**) 04 June 2019; (**d**) 25 June 2019; and (**e**) 13 July 2019. The accumulation of waste is displayed both qualitatively and quantitatively using a color gradation, which corresponded to the percentage of the surface area covered by the objects in relation to the total area. The sequence shows the pattern of spatial and temporal distribution, with a clear increase in waste accumulation in the upper part of the strips (the dune zone).

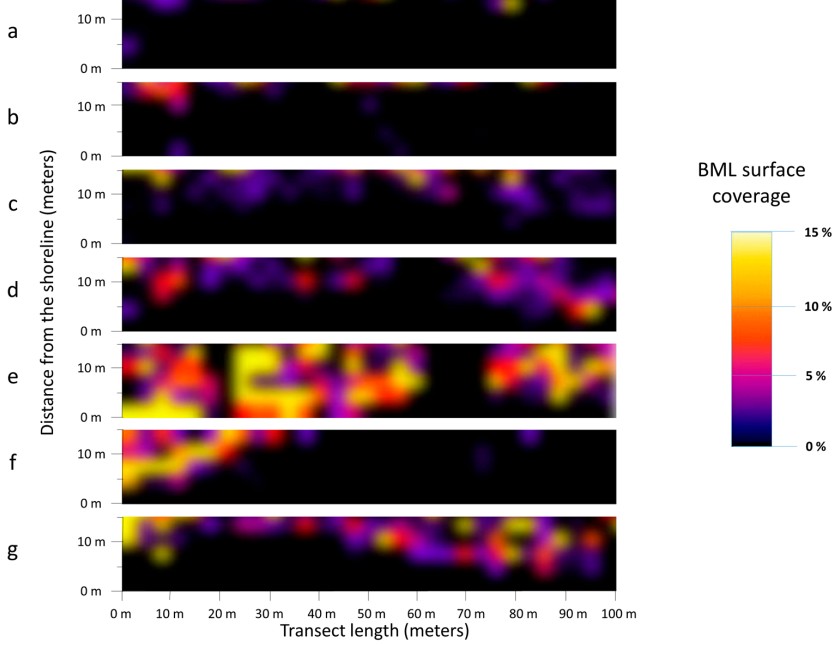

**Figure 9.** Variation in beached marine litter surface coverage of the monitored area, over time, for the second temporal period (from half July 2019 to January 2020). The correspondent dates were: (**a**) 19 July 2019; (**b**) 21 August 2019; (**c**) 18 September 2019, (**d**) 03 October 2019, (**e**) 20 November 2019, (**f**) 11 December 2019; and (**g**) 17 January 2020. As in Figure 8, the pattern of spatial and temporal distribution highlights the accumulation of waste increasing in the upper part of the strips (dune zones). In strip (**e**) the "footprint" of the Arno flooding, that occurred during the correspondent time period, is evident.

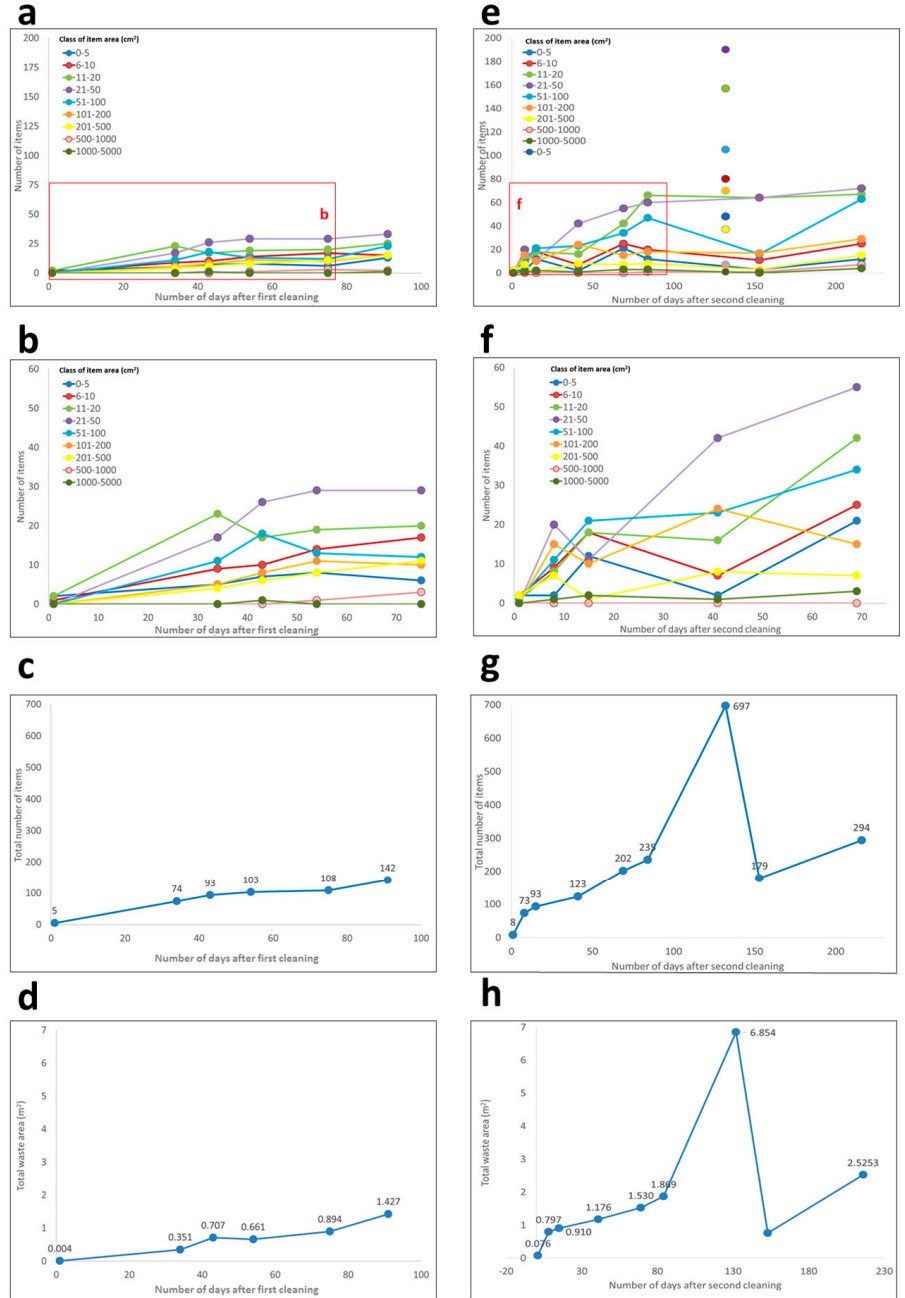

**Figure 10.** Accumulation trend of the BML with time for both time periods studied, starting from a "completely clean beach" situation, for the different size classes (**a**,**b**,**e**,**f**), in terms of the total number of elements accumulated (**c**,**g**) and the total area covered (**d**,**h**). The time, in the x- axes, for both periods starts from '0', which represents the date of total beach cleaning. The points correspond to the acquisition dates of the orthophotos (graphs on the left: 12 April 2019; 15 May 2019; 24 May 2019; 04 June 2019; 25 June 2019; 13 July 2019; graphs on the right: 13 July 2019; 19 July 2019; 03 August 2019, 21 August 2019; 18 August 2019, 03 September 2019, 20 November 2019, 11 December 2019, 17 January 2020). The "burst" detected on 20 November 2019, following the flood of the 15 November 2019 is well visible. Data collected before the first cleaning (12 April 2019) of the monitoring programme, that were not included in the graph, are the total number of items (203 items) and the total waste area (2.141 m$^2$).

Figures 8 and 9 show the distribution of the density (percentage of covered area) of beach waste in the studied area and its variation over time, from April 2019 to July 2019 and from July 2019 to January 2020, respectively. By using a color gradation to represent the amount of covered surface (see

Section 2.5), it was possible to highlight both qualitatively (visually) and quantitatively (percentage of covered surface) how the accumulation changed with time. In Figures 8 and 9, therefore, for both temporal periods, the initial "random" pattern of BML assumed, and with time, an increasingly precise connotation with a greater accumulation (red/yellow) mainly located near the dunes clearly emerges. The fact that in this type of beach most of the debris is in the area towards the dune was already highlighted by other studies, including those carried out using UAVs [43]. The new results that emerged from our long-term programme were the time-series that allowed us to obtain the trend of BML accumulation over three or four months, starting from a cleaned-up beach. In our opinion, these data are important because they can contribute to the understanding of the dynamic mechanisms that determine the "vertical" (cross-shore) distribution of BML observed on beaches.

*3.2. Quantity, Typology, and Accumulation Rate of BML*

Another important result obtained from our long-term monitoring programme is the estimation of the quantity of waste accumulated over time and its rate of accumulation, for different size classes of BML and for two different periods (spring–summer and summer–autumn).

In Figure 10, the graphs on the left side refer to the first period (spring–summer). Figure 10a,b show the number of objects deposited on the studied stretch of beach with time, for different size (surface area) classes (Figure 10b is just a zoomed-in view of Figure 10a). Figure 10c shows the total number of objects accumulated with time, while Figure 10d shows the total surface covered by BML, with time. Figure 10e–h show the same graphs but for the second period (summer–autumn). Looking at graphs (a)–(e), we note that, starting from the date of the total cleaning of the beach (date "0" for both periods), there was a progressive increase in waste on the beach for all size classes in the first months. Unfortunately, during the first period (spring–summer) we could not monitor frequently during the first month and, therefore, we did not have any data in the first 30 days, as can be seen in Figure 10b. On the contrary, the more frequent data-acquisition flights that were performed in the second period (summer–autumn) highlighted a clear general fast growth of items of all size classes in the first ten days (Figure 10f). Then, up to 40/50 days from the cleaning, this initial common steep growth changed to a less fast growth or, for some size classes, to a decrease. Finally, from the third month (around 60–80 days), the accumulation seemed to be in an "almost-flat" phase, with no or very low growth in the total quantity of items, for almost all size classes. The range of 10–40 days after the full beach cleaning was the one in which major changes in the accumulation occurred, depending on the size class. After 40 days, the growth continued, but at a much lesser rate, especially in the first period (spring–summer), both for the total number of objects and for the total area covered (Figure 10c,d). The second period (summer–autumn) was characterized, in its final phase, by an anomalous accumulation due to a flood in the Arno river, which occurred on 15 November 2019. During this extraordinary event, there was a large increase in the flow rate of the river, with a peak of 1473.75 $m^3$/s compared to the previous period (mean flow of 33.44 $m^3$/s, with a minimum of 8.7 $m^3$/s, and a maximum of 121.1 $m^3$/s). In addition, a strong south-west wind (Libeccio) was recorded, with a maximum peak of 86.4 Km/h and an average daily value of 50.1 Km/h, just in the direction of the coast [55]. Parallel to the increase in the transport of solids from the river and their discharge into the sea [20,45], the effect of the wind must also be taken into account, which prevents their dispersion offshore but, instead, helps to push them towards the coast where they accumulate on the beaches. Our monitoring, which took place on 20 November 2019, i.e., immediately after this flood, showed a huge increase in the number of stranded materials, highlighted both in Figure 10 (dotted line) and Figure 11, and through the high density and uniform spatial distribution of BML displayed in Figure 9e. In the following monitoring dates, we could observe a return to the pre-flooding values, both of the quantity and of the spatial distribution of the BML, as evidenced again in Figures 10 and 11 and in the last strip of Figure 9, corresponding to our last survey (17 January 2020). The data concerning the number of BML, obtained during this last survey (performed in January 2020), are reported in Figure 10 and included in Tables 1 and 2; from the Tables, and by comparing graphs (c) and (g) of

Figure 10, it can be noted that the amount of BML in the autumn and winter season was higher than that found in the previous seasons (spring–summer). The survey of 17 January 2020 thus concluded the second studied time-period (summer–autumn) with a considerable delay, as it was impossible for us, after the last monitoring on 11 December 2019, to carry out other surveys with UAVs, due to the continuous bad weather conditions, which lasted until mid-January 2020.

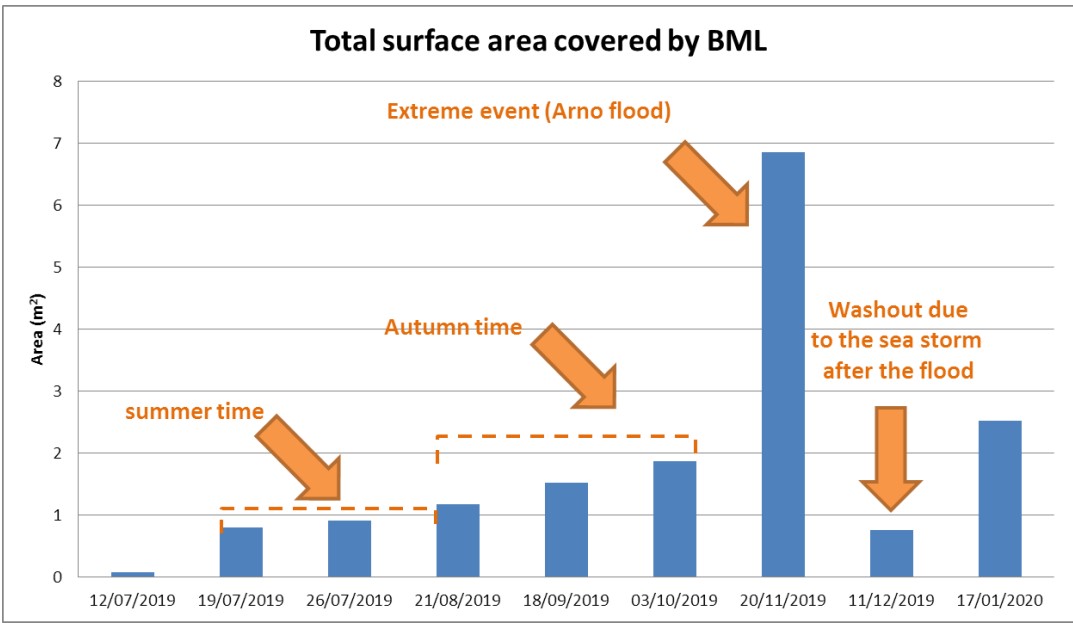

**Figure 11.** Histogram of the total surface area covered by BML for the different survey dates of the temporal period summer–autumn. Note that the time gap between different dates is not constant.

### 3.3. Comparison with "Standard" Survey Results

Standard monitoring campaigns were carried out during our long-term monitoring program with the help of volunteers, citizens, researchers, and students, following the SeaCleaner protocol [52,53] (which meets the MFSD survey procedures [16] and the OSPAR guideline for size and type classification of BML [15]). To date, three beach cleaning operations have been carried out—at the beginning of the first period (12 April 2019), at the beginning of the second period (13 July 2019), and at the beginning of the third, for which data acquisition has not yet been completed (17 January 2020). The cleaning and cataloguing operations take about one full day for each date of monitoring. Tables 1 and 2 show the results from the manual standard survey of the studied beach, compared with the one obtained from ortho-mosaic.

**Table 1.** Results from manual standard survey of the studied beach, compared with the one obtained from ortho-mosaic (material type classification).

| | 12 April 2019 | | | 13 July 2019 | | | 17 January 2020 | | |
|---|---|---|---|---|---|---|---|---|---|
| | Standard Survey | UAV Results | UAV vs.* Standard (in Percentage) | Standard Survey | UAV Results | UAV vs.* Standard (in Percentage) | Standard Survey | UAV Results | UAV vs.* Standard (in Percentage) |
| MATERIAL | Number of items | | | Number of items | | | Number of items | | |
| Plastic** | 879 | 182 | 20.71% | 741 | 142 | 19.16% | 1503 | 277 | 18.43% |
| multimaterial | 42 | 6 | 14.29% | 28 | 3 | 10.71% | 70 | 2 | 2.86% |
| Glass | 17 | 12 | 70.59% | 3 | 2 | 66.67% | 10 | 5 | 50.00% |
| Metal | 4 | 3 | 75.00% | 3 | 2 | 66.67% | 17 | 10 | 58.82% |
| Other (Clothes . . . .) | 1 | 0 | 0.00% | 2 | 2 | 100.00% | 1 | 0 | 0.00% |
| Total items | 943 | 203 | 21.53% | 768 | 151 | 19.66% | 1599 | 294 | 18.39% |
| | Density (items·/m$^2$) | | | Density (items·/m$^2$) | | | Density (items·/m$^2$) | | |
| Total density | 0.63 | 0.13 | 20.63% | 0.51 | 0.1 | 19.61% | 1.07 | 0.2 | 18.69% |

Note: *Percentage of litter identification compared to that derived using the standard terrain assessment. **including EPS, rubber, foam.

**Table 2.** Results from a manual standard survey of the studied beach, compared with the one obtained from ortho-mosaic (dimensional classification).

| Date | 12 April 2019 | | | 13 July 2019 | | | 17 January 2020 | | |
|---|---|---|---|---|---|---|---|---|---|
| | Standard Survey (Number of Items) | UAV Results (Number of Items) | UAV vs.* Standard (in Percentage) | Standard Survey | UAV Results (Number of Items) | UAV vs.* Standard (in Percentage) | Standard Survey | UAV Results (Number of Items) | UAV vs.* Standard (in Percentage) |
| Small (2.5–15 cm) | 859 | 124 | 14.44% | 716 | 103 | 14.39% | 1526 | 240 | 15.73% |
| Medium (15–50 cm) | 67 | 64 | 95.52% | 49 | 46 | 93.88% | 60 | 45 | 75.00% |
| Large (> 50 cm) | 17 | 15 | 88.24% | 3 | 2 | 66.67% | 13 | 9 | 69.23% |
| Total | 943 | 203 | 21.53% | 768 | 151 | 19.66% | 1599 | 294 | 18.39% |

Notes: *Percentage of litter identification compared to that derived using the standard terrain assessment.

In a 1500 m$^2$ transect on 12 April 2019, we detected a total of 943 objects through a standard manual census, while the visual screening of UAV orthophotos, for the same area, reported 203 objects. Thus, the percentage of litter identification, from images at an altitude of 6 m, compared to that derived using the standard terrain assessment, was about 21%. At the end of the first monitoring period, on 13 July 2019, 768 items with manual census and 151 items from UAV ortho-photos screening were counted. Therefore, the probability of litter identification compared to the standard manual census, was about 20%. At the end of the third monitoring period, on 17 January 2020, 1599 items with manual census and 294 items from UAV ortho-photos screening were counted—which gave an 18% probability of litter identification, compared to the standard manual census. From Table 2 we can see that the main differences between the standard manual and the UAV concern the small objects, this is not surprising and it agrees with what was observed in [43]. We must point out that the objects belonging to our "small" category had dimensions between 2.5–15 cm, in line with the OSPAR prescription. Although through the use of UAV and the other settings adopted, we would be able to detect even smaller objects (up to 1 cm of average linear dimension), during data extraction using the WM software we only counted BML included in the size of the "small" category, to be able to make the comparison with data extracted from the manual census.

In Table 1, the major differences between UAVs and the standards results were mainly found for the plastics and the multimaterial categories. This is because most plastics are small objects and multimaterials are inherently difficult to identify, as compared to other well-classified BML.

## 4. Discussion

The visualization of the orthographic maps allowed the study of the two-dimensional distribution of the accumulations (Figures 8 and 9), which was not feasible with the standard monitoring approach (manual collection and classification of the objects). The information we got in this way confirmed some previous results, such as the fact that BML accumulated prevalently in the dune zone [41,45], and was a cause for concern as, in this area, there were semi-permanent structures (trunks, clumps of plants, etc.) that contributed to retaining anthropogenic debris in the long-term, by hindering their return to the sea, even in conditions of heavy storms. This favored their photo-degradation, which was faster on the rather than the sea [8] and led to the consequent formation of meso and micro-plastics, [9]. However, with our monitoring approach we were also able to evaluate the dynamics of the accumulation process, and its dependence on the object sizes. In fact, Figure 10 shows that in the first ten days (the first time range) there was a fast growth of the number of objects of any sizes. This was understandable, because we started from a cleaned beach. Then, during the second time range (from 10 to about 60 days), the dynamics was more influenced by the size of the objects (Figure 10b and especially Figure 10f). Probably, the great variability that characterized this time interval was due to the different dynamic equilibrium times, between deposition and removal, for objects of different size classes. From the third month, the general trend was a regular growth with a dynamic that decreased a lot, for all size classes, towards the equilibrium between deposition and removal, possibly disturbed by the occurrence of sporadic events such as, in our case, the flooding of the river (Figure 10g,h). At the end of this last period, it was not unusual to see a large prevalence of small objects: this agreed with the results of previous manual surveys, which considered the size classes, carried out in the same area [53,56]; more generally this agrees with the fact that the number of macro-AMDs was higher when the size was smaller, both at sea and on the coast. [55,57].

The predominance of smaller objects led to some issues concerning the counting method using an aerial survey. In fact, despite the expected accuracy in counting beach objects (85%–100%, Figure 7) with UAV, for small objects, the figure obtained when compared to the standard (manual) counting, was quite different—about 15% (Section 3.3, Tables 1 and 2). This result agreed with that of Martin et al. [43]; in particular that result shown in Table 2 They pointed out that "smaller items < 4 cm in average linear dimension, were those that were not mainly recorded through aerial surveying (small fragments, bottle caps, plastic rings, etc.)". It is interesting to note that litter density estimates by

Martin et al. [43] through aerial survey and manual image processing (0.27 items/m2) was in line, even if slightly higher, with what we observed (Table 1). However, we must point out that their surveys were carried out in a very different place from ours, so this agreement between the two measures could only be fortuitous. As far as counts performed manually using the SeaCleaner protocol are concerned, the numerical density values (items/m$^2$) found (0.51–1.07, see Table 1) agreed well with those found with the same protocol during previous surveys, in the same geographical area [53,56].

In any case, our work supported the conclusion of others [42,43], regarding the difficulty in identifying the smallest objects through UAV. The reasons for this gap between manual and UAV counting of small objects could be the following:

1.  Hidden BML (for example under trunks or other objects) could be easily identified by human inspection, while this was almost impossible for UAV;
2.  Almost completely buried BML could be extracted by humans from sand, identified and then counted, while UAVs cannot do the same, obviously;
3.  Some transparent BML, especially fragments of plastic bags and thin films, are often not detected by the UAV camera;
4.  Small BMLs can be overestimated by manual counting. In fact, while for the protocol suggested by the OSPAR macro-waste guidelines (to which our specific SeaCleaner local monitoring protocol refers), objects smaller than 2.5 cm should not be considered [15,52,53,55], however, during manual counting such small objects were often counted equally, contrary to the recognition and counting by orthophotos.

In the course of this work, we have noted the importance of taking into account the occurrence of extreme events, as evidenced by the "burst" in Figures 10 and 11, where the increase in the flow river heavily affected the transport of the solid bodies by the rivers, and so influenced the accumulation of BML [20]. It is interesting to note that, after this extreme event, the situation reverted back to that similar to previous ones (Figures 10–12). The decrease in detected BML after the flood event was due to the successive swells, i.e., the dynamics of the waves, which, even in "standard" conditions, not only tend to accumulate, but also remove objects from the shoreline, tending towards a situation of equilibrium. As a result of this, in normal conditions most of the waste is found in the dune area, where the presence of trunks and plant material tend to prevent it from returning to the sea (Figure 12a). During exceptional events, such as that of 15 November 2019 (Figure 12b), the enormous amount of material transported and washed up on the beach produced situations such as that in Figure 12c, i.e., a significant increase in the number of waste, evenly distributed on the beach (Figure 10e). The waves (and even more so the strong swells) that act in the period following this event, partly removed this material, bringing the beach back to a situation similar to the previous ones with regards to the number and density of BML, with the trash mainly accumulated in the dune area (Figures 9g and 11). In our particular case, the heavy swells following the flood also contributed to a change in the conformation of the beach, removing a part of the sand from the ordinary berm zone (the upper part of the beach, with depositional features due to the accumulation of sediment caused by the waves, Figure 12d). Moreover, the removal time of the plant material accumulated in the dune area was quite long—it was in fact present, in a higher than normal quantity, even during our last monitoring, which was 32 days after the occurrence of the exceptional event when the beach had recovered its standard conformation. This had probably caused a greater difficulty in detecting the BML with aerial devices, with even relatively large BML sometimes being hidden from aerial detection (see Point 1 above), as observed in our latest census that showed a worse agreement between UAV and standard manual survey, also for "large" and, especially, for "medium" objects, with respect to previous dates. A part of this small difference, the percentage of BML identification with UAV compared to that derived using the standard terrain assessment was very similar for all three dates reported in Tables 1 and 2, ranging from 18.39% to 21.53% for total item number, and from 18.69% to 20.63% for total density of items, indicating that the bias described above (points 1, 2, and 3) always affect in the same way. However, our study was limited

to a small stretch of beach (100 meters long and 15 meters deep), while extreme events also affected the shape of all coastlines of SRPRK. Therefore, an effective understanding of the consequences of these strong events should imply the study of a large portion of the coastline, also taking into account the erosion phenomena.

Even if not quantitatively reported in this work, we have observed an effective role of the wind influencing the special distribution of BML. This applies, in particular, to expanded polystyrene (EPS), which is easily fragmented and, because of its very low density, it was conveyed by the wind more than by the sea. Not surprisingly, from Figure 10 and also Tables 1 and 2, we noted that the amount of BML accumulated during spring and summer was smaller than that during late autumn and in the winter, which were characterized by stronger winds and larger swells.

The main task of the present work was to test a possible methodology for studying the coastal dynamics of waste accumulation through aerial survey devices. Thus, our attention was more focused on the study of accumulation dynamics in the coastal area rather than on the precision and detailed cataloguing of the found objects, as done by other works [43,44]. However, we realized that the high precision in object recognition could lead to a high accuracy even for its size (surface area) estimation, a priority target for understanding the dynamic behavior of different classes of AMDs on the beaches (Figure 10). Actually, the aerial survey size evaluation was intrinsically influenced by some errors that led to an underestimation of the values. For example, as discussed in Point 2 of the list above, and due to the fact that the area of a "flat" BML positioned in an almost vertical direction was highly underestimated by UAVs, because of the almost fixed direction of view. The estimation of this bias required a dedicated study.

Aerial surveys save time, compared to standard manual approaches to the BML study, even after accounting for the time needed, in addition to mere monitoring, for image processing, labeling, and imaging. In fact, covering even larger areas requires the work of only one person, and data extraction by orthophotos requires no more than 2 hours of visual census, for each monitoring campaign. Moreover, geo-referenced images can provide useful information that standard counting cannot provide. The study of accumulation dynamics, like ours, requires two/three monthly monitoring campaigns, repeated for 3/4 months. Many people should be involved in standard manual procedures. This is probably the reason why data on the accumulation rate of BML are scarce in the literature. The use of UAVs can, in our opinion, help to fill this gap. In addition to the pure scientific aspect of the phenomenon, the knowledge of the accumulation behavior, possibly in different areas of the coast, was a useful information for marine parks/protected areas (MPAs). Presently UAV technology, compared to some years ago, is sufficiently low-cost and it can be foreseen that parks and administrations of MPAs are equipped with such devices, with the patents/permits related to their use. The knowledge of the time scale for the formation of the maximum stock of BML after the beach cleaning operation, could help the MPAs and the local authorities to optimize the planning of the cleaning campaigns, minimize the effort, and maximize the result, thus, preventing the degradation and fragmentation of most of the material and the production of microplastics.

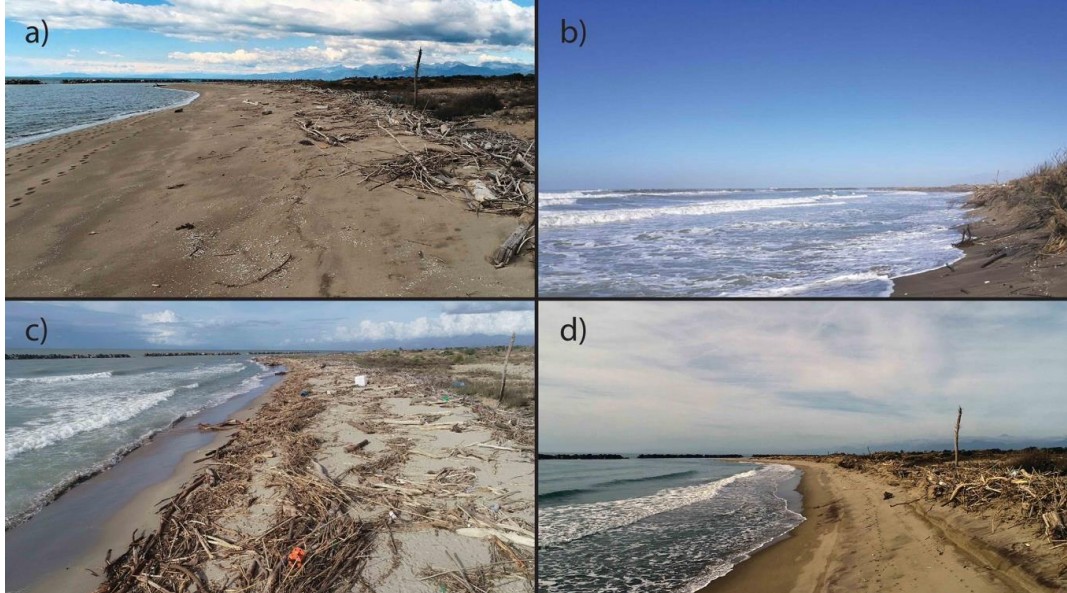

**Figure 12.** (**a**) The beach before the sea storm of 20/11/2019 (normal conditions). (**b**) The beach during the storm. (**c**) The beach 3 days after the storm. (**d**) The beach 10 days after the storm. At the time of the last monitoring (17 January 20120, i.e., 32 days after the storm) the beach was returned almost to its normal condition and conformation.

## 5. Conclusions and Further Improvement

The UAVs flying at a low altitude provided high resolution data, which was useful in detecting plastic, metal, and other type of beached objects. Moreover, UAV allowed the repeatability of the surveys in a short time, which was essential for the study of accumulation dynamics. Other important advantages were the reduced anthropogenic impact (just one person for the survey) and the possibility of obtaining 3D and 2D characterization of the monitored areas. Our pilot test to use UAVs for monitoring spatial and temporal dynamic of BML accumulation in coastal areas that started in April 2019, proved to be a useful procedure. To our knowledge, this was the first case of using the aerial survey methodology through UAVs to monitor the presence of BML on Italian beaches (Figure 7 of [40]) and the first case of using UAVs to estimate the accumulation rates of BML on beaches in general. The results of our work showed that the dynamics and the equilibrium of the accumulation process depended, in general, on season, but also on the size of the specific BML. Moreover, extreme events lead to strong fluctuations, but the normal situation was quickly restored, for both the dynamics and the equilibrium features. Observational evidence of this phenomenon, as well as that visible in the peak accumulation of BML recorded by us (Figures 9–11), are available in our short film "Before&3-10dayafter.avi" (Supplementary Material), which shows the situation of the beach before the flood, three days after this extreme event and, finally, ten days after it.

The comparison between the UAV and standard manual counting (made according to the SeaCleaner protocol [52,53], which met the MFSD survey procedures [16] and the OSPAR guideline for size and type classification of BML [15]) showed a good agreement for "medium" and "large" size objects (~67%–95%), while this was not the case for the "small" ones (~15%). The possible causes of this discrepancy are analysed in the Discussion section.

Our study focused on the distribution of BML in the different areas of a stretch of beach, from the coastline to the dunes. In a next work, we would like to analyze, with the same techniques, the dynamics of accumulation on several stretches of beach of the entire coastline of the park. In this way, we could provide information on the importance of both, the distance from the mouths of rivers (Arno and Serchio) and the presence of possible obstacles in the process of accumulation of debris. To this end, we are currently testing different types of UAVs and increasing the flight height (requesting the

necessary clearance in advance). The objective was to increase the size of the scanned area for each individual flight, losing a little of the detail, in the process. Ideally, we should do a lower resolution scan on the whole coastal area of the park, and do a detailed analysis, like the one in this paper, on a few randomly sampled areas.

As shown in Tables 1 and 2, besides the fact that UAV census counts are generally underestimated for all categories of objects, it seemed that the recognition of some particular types was more difficult and in our case was the "multimaterial". As already pointed out in the Discussion section, although the recognition of the object typology through UAV did not play a central role in the present work, its importance in improving the effectiveness of the monitoring system (estimation of the count and the surface/volume of the objects) had emerged. Therefore, we are trying to apply automatic systems based on machine learning, as already tested by other BML recognition studies [40,41,43,44]. In order to carry out a more accurate study of how coastal dynamics affect the accumulation of BML, it would also be necessary to cross-reference the data obtained from AMD monitoring campaigns with those related to sea and wind weather conditions, while accounting for extreme events, as far as possible.

The main difficulties encountered in this type of monitoring are, in our opinion and personal experience, those common in autumn and winter due to adverse weather conditions; windy or rainy days or heavy swells prevent the surveys from being carried out (see Figure 12) and might, therefore, compromise the planned surveys (as for our last late survey of the summer–autumn period). In particular, the flight operation must be suspended if it rains, the wind exceeds 10 m/sec and the temperature exceeds the range of 0–40 °C. Operating near the shoreline, the force of the sea can produce aerosol in the area–low aerosol levels can be managed with a simple cleaning, while high values (present with rough seas and strong wind conditions) can damage the UAV and require extraordinary assistance. Take-off and landing can raise the sand, which can damage the moving parts; it is therefore, important to use an appropriate drone landing pad. Even sub-optimal light conditions (Figure 7) and ground shading can affect the results–during the scan it is preferable to have a constant brightness, therefore, the fast passage of clouds could produce variations of light that needs to be corrected in post-processing. Another operational limit is the flight autonomy of the UAV, especially if one wants to analyze large areas; in this case the APP Drone Harmony was very useful, because it managed the interruption of the flight, the replacement of the battery, and the resumption of the flight starting from the last position.

To date, the use of UAVs for BML monitoring had only just begun, and we think this work could help to highlight its great potential. It is, therefore, foreseeable that UAVs would be widely used in the future and would allow us to considerably increase the knowledge of the dynamics of accumulation of BML on beaches, especially in coastal areas with difficult access and MPAs. The understanding of the characteristics of this process and the possibility for acquiring a large amount of data, even in real-time, combined with the relatively modest costs of these methodologies, would help allow, through integrated programs, the different stakeholders involved in mitigation actions, citizencience, and research activities, to work together to optimize beach cleaning actions.

**Supplementary Materials:** The following are available online at http://www.mdpi.com/2072-4292/12/8/1260/s1, Figure S1: The figure summarizes the potential of the three-dimensional orthophoto reconstruction. In (**a**) a relatively large section of the beach is shown, with the largest BML in evidence, while in (**b**) a smaller area of the selected beach section has been enlarged. The resolution of this method is high enough to identify even small objects in a large area.

**Author Contributions:** Contribution by different authors can be described as follows: Conceptualization S.M. and M.P.; methodology: S.M., M.P. and L.M.; software: M.P. and L.M.; validation: S.M., M.P. and L.M.; formal analysis: M.P. and L.M.; investigation: S.M. and M.P.; resources: A.B. and M.P.; data curation: M.P.; writing—original draft preparation: S.M.; writing—review and editing: S.M. and L.M.; visualization: M.P. and L.M.; supervision: A.B. and L.M.; All authors have read and agreed to the published version of the manuscript.

**Funding:** This research received no external funding.

**Acknowledgments:** We would like to thank the environmental associations "Acchiapparifiuti" and "Legambiente" (www.legambiente.it), and the working group Win on Waste (WOW—interdepartmental group of the National Research Council) for the support given to us during the manual operations of cleaning and classification of

waste, as well as for the outreach activities carried out in order to further spread awareness of the problem of Marine Litter, with the production of video and educational material (UAVSanRossoreENG.m4v, Supplementary Material).

**Conflicts of Interest:** The authors declare no conflict of interest.

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
