# Peer review of "Unmanned Aerial Vehicles for Debris Survey in Coastal Areas: Long-Term Monitoring Programme to Study Spatial and Temporal Accumulation of the Dynamics of Beached Marine Litter"

_remotesensing, doi:10.3390/rs12081260_

Round 1

Reviewer 1 Report

General comments

More details on the software, the semi-automatic algorithm and the establishment of the training dataset should be given. It is not possible to reproduce this study without more details.

Consider merging Results and Discussion, as in the current version results also include interpretation and discussion.

Can you include details on operational limitations? For example flying under windy conditions?

Specific comments

Abstract:

Include duration of long-term monitoring programme

Include location of monitoring programme

1. Introduction

Line 35: Litters --> Litter

Line 43 – 53: Include recent work by Olivelli et al. (2020) on the important sink that coastlines seem to play in the overall mass budget of plastic pollution.

Line 52: You can emphasize that rivers seem to play a key role in the transport of plastic debris from land to ocean (van Emmerik & Schwarz, 2020).

Line 70 – 75: Include the recent work by Geraeds et al. (2019) who used UAVs for the first time to monitor floating plastics in rivers.

Line 98: Please expand on the general concept of the software used.

Line 101: Include how much area was covered each flight.

2. Material and methods

Figure 1: Please improve the quality of the figure. Include (1) legend, (2) north arrow, (3) scale, and (4) length indication of the sampling site. Consider using specific software to make maps (e.g. ArcGIS, QGIS). Also please indicate where the rivers are located.

Line 112: What does “limitation” mean? How many tourist visit this area daily/annually?

Line 121: Deep --> wide?

Footnote 1: Please include this in the main text.

Line 130: Define good resolution.

Figure 2: Maybe include the meaning of the variables in the legend, in addition to the caption.

Line 154: Include source. How many pixels per item did you assume to be sufficient?

Line 165: What was the overlap percentage based on?

Line 175: How were the flight plans memorized? Did you use an app like Litchi?

Line 177: Removal up to what size?

Line 186: What framework did you use for classification and counting?

Line 197: Is the software available to others?

Figure 5: Please use a different colormap than rainbow, as it is still considered harmful. Also please label the axis.

Line 444 – 446: Their surveys were done at a very different location. I think this is mere luck.

Figure 12: Present this in the results, not in discussion.

Line 481: How much time is saved? Also Remote sensing imagery needs to be processed, labeled and trained. Is there really time saved?

Line 500: How extreme were the events?

Line 526: Please include observational evidence for the extreme events that you considered.

3. Results

Line 271: Please provide a range for “few hours”.

Line 280 – 285: Move this to the methods.

Figure 8, 9: Please add labels to and titles to the figures.

Figure 1-: Why is there a different color for Day 132 for subplots g and h? Why don’t you include the data from the surveys after the first cleaning?

Line 346: Can you include more details of the flood event? What was the discharge? What kind of return period? Increased discharge may indeed lead to more plastic transport from rivers (van Emmerik et al., 2019), but you should make this link clearer in the text.

Line 359: D map --> 3D map?

Line 367 – 381: What is the use of the 3D images if you are not using them? I suggest to either remove this, or include additional analysis on the 3D images. Can you estimate volume or something?

Table 1, 2: Please include the units of the numbers at the top of the columns. Also, by dividing UAV by Standard I expect that you present a factor, not a percentage. Can clarify this unit?

Line 411: Can you indicate somewhere what the minimum item size is that you detected? What about only using the OSPAR data with item categories that is equal or larger than the UAV minimum item size?

References

Borland, D., & Ii, R. M. T. (2007). Rainbow color map (still) considered harmful. IEEE computer graphics and applications27(2), 14-17.

Geraeds, M., van Emmerik, T., de Vries, R., & bin Ab Razak, M. S. (2019). Riverine plastic litter monitoring using unmanned aerial vehicles (UAVs). Remote Sensing11(17), 2045.

Olivelli, A., Hardesty, D., & Wilcox, C. (2020). Coastal margins and backshores represent a major sink for marine debris: insights from a continental-scale analysis. Environmental Research Letters.

Author Response

We would like to warn the revisors and the editor that, due to a malfunction of our “Word”, and in the impossibility of being able to access our office to remedy the problem quickly, we were forced to number the lines page by page, instead of using the option of progressive numbering, as this one worked very badly and caused problems to the readability of the document. So, our references to the corrections made will be given with both the page number and the line/lines number on the indicated page the figures in the manuscript are a reduced and less defined version of the ones sent in the zipped file (especially the Fig. 8 and 9), to avoid problems in the transmission of the documents, having to send everything from home and not having a very good line.

General comments

R Point 1 More details on the software, the semi-automatic algorithm and the establishment of the training dataset should be given. It is not possible to reproduce this study without more details.

A Response 1 We developed this part better, from line 13 of pag. 8  to line 14 of pag.9

R Point 2Consider merging Results and Discussion, as in the current version results also include interpretation and discussion.

A Response 2 Having developed some points of the "discussion" paragraph in more detail, following the requests of the other reviewers, we thought it was preferable to keep the two parts separated. We hope these changes will serve to clarify, better than before, both sections of the manuscript.

R Point 3 Can you include details on operational limitations? For example flying under windy conditions?

A Response 3 It was done from the line 44 to line 49 of pag. 20.

Specific comments

Abstract:

R Point 4 Include duration of long-term monitoring programme

A Response 4 See line 21 pag. 1

R Point 5 Include location of monitoring programme

A Response 5 See line 20-21 pag 1

  1. Introduction

R Point 6 Line 35: Litters --> Litter

A Response 6 We corrected it, see line 36 pag 1

R Point 7 Line 43 – 53: Include recent work by Olivelli et al. (2020) on the important sink that coastlines seem to play in the overall mass budget of plastic pollution.

A Response 7 We have included the reference indicated, see line 4, pag 2, and updated the bibliography

R Point 8 Line 52: You can emphasize that rivers seem to play a key role in the transport of plastic debris from land to ocean (van Emmerik & Schwarz, 2020).

A Response 8 We have emphasized this concept, and included the reference indicated, see lines 10-11 of pag. 2, and updated the bibliography

R Point 9 Line 70 – 75: Include the recent work by Geraeds et al. (2019) who used UAVs for the first time to monitor floating plastics in rivers.

A Response 9 We have included the reference indicated, see line 33 of pag 2, and updated the bibliography

R Point 10 Line 98: Please expand on the general concept of the software used.

A Response 10 In the Materials and Methods paragraph we have added a paragraph specifically dedicated to the description of this software; so here, at the lines 7-8 of pag. 3, we have made a "reference" to this part described below.

R Point 11 Line 101: Include how much area was covered each flight.

A Response 11 See line 11-13 of pag. 3

  1. Material and methods

R Point 12 Figure 1: Please improve the quality of the figure. Include (1) legend, (2) north arrow, (3) scale, and (4) length indication of the sampling site. Consider using specific software to make maps (e.g. ArcGIS, QGIS). Also please indicate where the rivers are located.

A Response 12 See new Fig 1 included. In the manuscript, a reduced and less defined version than the one prepared was inserted, to avoid problems in the transmission of the documents, having to send everything from home and not having a very good line. However, all the revised images in high definition were sent to the magazine.

R Point 13 Line 112: What does “limitation” mean? How many tourist visit this area daily/annually?

A Response 13 See from line 3 to line  7 of pag. 4

R Point 14 Line 121: Deep --> wide?

A Response 14 We corrected it, see line 13 of pag 4.

R Point 15 Footnote 1: Please include this in the main text.

A Response 15 See line 29-31 of pag. 3

R Point 16 Line 130: Define good resolution.

A Response 16 See lines 18 - 20 of pag. 4

R Point 17 Figure 2: Maybe include the meaning of the variables in the legend, in addition to the caption.

A Response 17 See new Fig 2 included

R Point 18 Line 154: Include source. How many pixels per item did you assume to be sufficient?

A Response 18 See from line 40 to line 44 of pag 4.

R Point 19 Line 165: What was the overlap percentage based on?

A Response 19 See from line 17  to line 21 of pag. 5

R Point 20 Line 175: How were the flight plans memorized? Did you use an app like Litchi?

A Response 20 See from line 4 to line 7  of pag 6

R Point 21 Line 177: Removal up to what size?

A Response 21See from line 8 to line 10  of pag 6

R Point 22 Line 186: What framework did you use for classification and counting?

A Response 22 See from line 11 to line 16  of pag 6

R Point 23 Line 197: Is the software available to others?

A Response 23 See from line 31 to line 33  of pag 6 and note number 1

R Point 24 Figure 5: Please use a different color map than rainbow, as it is still considered harmful. Also please label the axis.

A Response 24 See new Fig 5 included

R Point 25 Line 444 – 446: Their surveys were done at a very different location. I think this is mere luck.

A Response 25 We have included this consideration in the text (see lines 38-42 of pag. 17)

R Point 26 Figure 12: Present this in the results, not in discussion.

A Response 26 Now it is in results paragraph. Fig 11.

R Point 27 Line 481: How much time is saved? Also Remote sensing imagery needs to be processed, labeled and trained. Is there really time saved?

A Response 27 We briefly include this consideration, from line 5 to line 9 of pag. 19 In any case, we confirm that this is a time-saving method, especially for the purpose we have set ourselves, i.e. to study the behaviour of accumulation over long periods of time and with monitoring at short distances from each other. In the "standard" way, i.e. with manual surveys, carried out every 10/15 days, it would not only have been burdensome from the point of view of the time to be spent, but also because of the greater number of people to be employed

R Point 28 Line 500: How extreme were the events?

A Response 28 See from line 22 to line 29 of pag. 14

R Point 29 Line 526: Please include observational evidence for the extreme events that you considered.

A Response 29 See from line 11 to line 14 of pag. 20

  1. Results

R Point 30 Line 271: Please provide a range for “few hours”.

A Response 30 See  line 24 of pag. 10

R Point 31 Line 280 – 285: Move this to the methods.

A Response 31 Now it is in material and methods paragraph. See from line 14 to line 19 of pag. 9

R Point 32 Figure 8, 9: Please add labels to and titles to the figures.

A Response 32 Labels were already on the figures; we added title to the legend

R Point 33 Figure 1-: Why is there a different color for Day 132 for subplots g and h? Why don’t you include the data from the surveys after the first cleaning?

A Response 33 See the changes inserted in the caption of Figure 10.  We thought to highlight the consequence of the exceptional event detected, and therefore the occurred "burst", separating it from the growth curve. In this way it is also clearer what the trend of this curve towards equilibrium is, regardless of the "contribution" of the exceptional event in question. We have not included in the graph the values of total items and density of BML present in the beach before our first cleaning, because the graph 10 wants to measure the growth curve starting from a clean beach situation. In any case, we have now included them in the caption of the figure in question.

R Point 34  Line 346: Can you include more details of the flood event? What was the discharge? What kind of return period? Increased discharge may indeed lead to more plastic transport from rivers (van Emmerik et al., 2019), but you should make this link clearer in the text.

A Response 34 See from line 22 to line 31 of pag. 14. We also quote “van Emmerik et al., 2020” and “Geraeds et al. 2019)

R Point 35 Line 359: D map --> 3D map?

A Response 35 We corrected the oversight. In any case, this part has now been excluded from the main text of the manuscript, and moved to Supplementary Material

R Point 36 Line 367 – 381: What is the use of the 3D images if you are not using them? I suggest to either remove this, or include additional analysis on the 3D images. Can you estimate volume or something?

A Response 36 This part has now been excluded from the main text of the manuscript, and moved to Supplementary Material. We hope to have, in the future, the possibility to use it, and so to give estimation of volume of beached objects. For now, we have not yet addressed this, but we believe it to be possible; in that case, as suggested by [43], it would be possible  to give a mass estimate of the accumulated material, too.

R Point 37 Table 1, 2: Please include the units of the numbers at the top of the columns. Also, by dividing UAV by Standard I expect that you present a factor, not a percentage. Can clarify this unit?

A Response 37 See new Table 1 and 2 included

R Point 38 Line 411: Can you indicate somewhere what the minimum item size is that you detected? What about only using the OSPAR data with item categories that is equal or larger than the UAV minimum item size? 

A Response 38 See from line 18 to line 21 of pag. 16 more line 1 and 2 of pag. 17.

Suggested References

R  van Emmerik, T. & Schwarz, A. Plastic debris in rivers. WIREs WATER, 2020, V7, Issue 1, https://doi.org/10.1002/wat2.1398

A Response Now it is reference number 20

R Geraeds, M., van Emmerik, T., de Vries, R., & bin Ab Razak, M. S. (2019). Riverine plastic litter monitoring using unmanned aerial vehicles (UAVs). Remote Sensing11(17), 2045.

A Response Now it is reference number 45

R Olivelli, A., Hardesty, D., & Wilcox, C. (2020). Coastal margins and backshores represent a major sink for marine debris: insights from a continental-scale analysis. Environmental Research Letters.

A Response  Now it is reference number 12

Please see also attached file

Reviewer 2 Report

The article is interesting from a scientific point of view and for a wide range of researchers who use low-level photogrammetry methods. The authors point out the urgent need to develop new methods of spatial and temporal mapping of beaches to identify the areas of greatest accumulation, quantify the abundance and types of material, and trace their origin, in line with the protocol and standard monitoring strategies. UAVs have been used to acquire geo-referenced RGB images in a selected zone of a protected marine area, during a long-term monitoring programme.

The photogrammetric flight was well designed with good parameters of aerial images: “specific precautions have been used to ensure optimal image capture: 1 m/s speed with "stop and go" mode for each shot to ensure shooting in a stationary position due to low flight altitude and to avoid blurred photos; manual focus set to infinity (i.e. autofocus disabled) to avoid variations in focus; initial exposure setting so that autoexposure (AE) is disabled to avoid variations in brightness”.

In the Discussion chapter in future publications should be explained more clearly: does the decrease in the amount of BML over time result from a smaller amount of their incoming or a decrease in their recognition rate by the WM system?

Author Response

We would like to warn the revisors and the editor that, due to a malfunction of our “Word”, and in the impossibility of being able to access our office to remedy the problem quickly, we were forced to number the lines page by page, instead of using the option of progressive numbering, as this one worked very badly and caused problems to the readability of the document. So, our references to the corrections made will be given with both the page number and the line/lines number on the indicated page the figures in the manuscript are a reduced and less defined version of the ones sent in the zipped file (especially the Fig. 8 and 9), to avoid problems in the transmission of the documents, having to send everything from home and not having a very good line.

General comments

R Point 1 In the Discussion chapter in future publications should be explained more clearly: does the decrease in the amount of BML over time result from a smaller amount of their incoming or a decrease in their recognition rate by the WM system?

A Response 1 We hope to have correctly understood the editor's comment (and therefore to have answered his questions) by expanding part of the "discussion", moving some parts of it and, above all, adding a whole paragraph from line 6 to line 37 of pag 18, with the intention of specifying and clarifying part of the results obtained, which probably in the previous version had not been rendered exhaustively.

see also attached file

Reviewer 3 Report

Dear authors,

Nevertheless the work is well done and structured, the novelty and originality of contents are really low. You talk about the UAV techniques as it was your first time, but the methodology you developed is really common. 

So any significant scientific contribution has been made.

I think that for the goals of the research it would be sufficient to use a camera design project.

When you talk about the GSD value, you report an image that is very used in the literature but the researchers of photogrammetric field don't use it but give the final results. 

Author Response

We would like to warn the revisors and the editor that, due to a malfunction of our “Word”, and in the impossibility of being able to access our office to remedy the problem quickly, we were forced to number the lines page by page, instead of using the option of progressive numbering, as this one worked very badly and caused problems to the readability of the document. So, our references to the corrections made will be given with both the page number and the line/lines number on the indicated page the figures in the manuscript are a reduced and less defined version of the ones sent in the zipped file (especially the Fig. 8 and 9), to avoid problems in the transmission of the documents, having to send everything from home and not having a very good line.

Comments

R POINT1: Nevertheless the work is well done and structured, the novelty and originality of contents are really low. You talk about the UAV techniques as it was your first time, but the methodology you developed is really common. 

So any significant scientific contribution has been made.

I think that for the goals of the research it would be sufficient to use a camera design project.

When you talk about the GSD value, you report an image that is very used in the literature but the researchers of photogrammetric field don't use it but give the final results. 

A RESPONSE  We do not claim that this is the first time that this methodology has been used. In fact, we report cases of monitoring in similar situations. In addition, we report all the references we know about the use of UAVs for marine litter monitoring. We only state that, to our knowledge, they have not yet been used for an estimate of the accumulation dynamics of such objects, but only for counting and recognition of their type.

In any case, we thank you for the positive comments on the structure of our pilot project, and to give us a chance that it will be published anyway.

Reviewer 4 Report

Some wording may be to change, such as "Anthropogenic Marine Debris". the marine debris itself is of anthropogenic origin. perhaps it is better to mention sources- terrestrial- rivers and beaches, shipping and waste dumping into the sea, ocean and coastal areas. some of the marine debris is carried by wind. 

Revise/rewrite this sentence - "The recent use of UAVs for low altitude monitoring of BML has made it possible to go beyond the usual low resolution limit of remote sensing by aerial systems (planes, satellites etc.)."- it is can be this way- "The UAVs flying at a low altitude provided high resolution data which was useful in detecting plastic, metal, etc objects.. "

To remove term of "remote sensing" from entire text and to mention only aerial survey, since there is no comparison with remote sensed data (acquired by satellite). It is not quite clear why satellite- remote sensing is mentioned here. It is understandable that RS data in this context have a low resolution, therefore it is not necessary to mention it here.

To provide a high resolution map of Italy- Figure 1 because it is not clear which province was selected for this study. To add province name, dots for capital- Rome and provincial centers.

Author Response

Comments

R POINT 1Some wording may be to change, such as "Anthropogenic Marine Debris". the marine debris itself is of anthropogenic origin. perhaps it is better to mention sources- terrestrial- rivers and beaches, shipping and waste dumping into the sea, ocean and coastal areas. some of the marine debris is carried by wind.  

A RESPONSE 1 Although the revisor's objection is correct, we prefer to use "Anthropogenic Marine Debris - AMD" because it is frequently used in the field of study in question. The studies of monitoring and quantification of anthropogenic objects (not only plastic, but any material) all use this definition, and also the acronym AMD is almost widely used by those who work in this field, being present in search engines.

As regard the principal sources of these objects, we partly stress the role of the rivers (pag.2, lines 9-11; pag 18 lines 6-9;), and partially the one of the wind (pag 14 lines 25-27; pag 18 line 38-43 and pag 20 lines 37-40). In this paper, we dedicated mostly to the study of the accumulation process on the beaches, carried on by waves. This process could be also enhanced by the wind (pag 14 lines 25-27) but to consider all these causes need a more deep study, that we hope to have the possibility to do in the future (pag 20 lines 20-29). The “source problem”, so, i.e. to investigate which are the principal sources of waste into the sea, requires other kind of approach which is beyond the scope of this work.

R POINT 2 Revise/rewrite this sentence - "The recent use of UAVs for low altitude monitoring of BML has made it possible to go beyond the usual low resolution limit of remote sensing by aerial systems (planes, satellites etc.)."- it is can be this way- "The UAVs flying at a low altitude provided high resolution data which was useful in detecting plastic, metal, etc objects.. "

A RESPONSE 2 We replaced the sentence. See pag. 19 line 32-33

R POINT 3 To remove term of "remote sensing" from entire text and to mention only aerial survey, since there is no comparison with remote sensed data (acquired by satellite). It is not quite clear why satellite- remote sensing is mentioned here. It is understandable that RS data in this context have a low resolution, therefore it is not necessary to mention it here.

A RESPONSE 3 We did it in all the text.

R POINT 4 To provide a high resolution map of Italy- Figure 1 because it is not clear which province was selected for this study. To add province name, dots for capital- Rome and provincial centers.

A RESPONSE 4 We did it. See new Fig 1. The one inserted in the manuscript is not in high definition for sending problems, but we sent it to editor in a zipped file.

see also attached file

Round 2

Reviewer 1 Report

Some final comments:

  1. Label the axes in Fig. 8 and 9.
  2. Add what BML stands for in caption
  3. In Fig. 10, please don’t use a different line for day 132. It’s fundamentally wrong to show it like this, as the lines are connected to data points on others days. The peaks are clear enough, further emphasis can be given in the text.

Author Response

Point 1: Label the axes in Fig. 8 and 9.

RESPONSE 1: See new Figures 8 and 9

Point 2: Add what BML stands for in caption

RESPONSE 1: See new captions of Figures 8 and 9

Point 3: In Fig. 10, please don’t use a different line for day 132. It’s fundamentally wrong to show it like this, as the lines are connected to data points on others days. The peaks are clear enough, further emphasis can be given in the text.

RESPONSE 3: see new figure 10 and corresponding caption

Reviewer 3 Report

The improvement that the authors have been made allow the pubblication of the paper.

Author Response

Thanks for all the suggestions and comments.